# The N-terminal Subunit of the Porcine Deltacoronavirus Spike Recombinant Protein (S1) Does Not Serologically Cross-react with Other Porcine Coronaviruses

**DOI:** 10.3390/pathogens11080910

**Published:** 2022-08-13

**Authors:** Lu Yen, Ronaldo Magtoto, Juan Carlos Mora-Díaz, Jose Antonio Carrillo-Ávila, Jianqiang Zhang, Ting-Yu Cheng, Precy Magtoto, Rahul K. Nelli, David H. Baum, Jeffrey J. Zimmerman, Luis G. Giménez-Lirola

**Affiliations:** 1Department of Veterinary Diagnostic and Production Animal Medicine, College of Veterinary Medicine, Iowa State University, Ames, IA 50011, USA; 2Andalusian Public Health System Biobank, 18016 Granada, Spain; 3Department of Veterinary Preventive Medicine, College of Veterinary Medicine, Ohio State University, Columbus, OH 43210, USA; 4College of Veterinary Medicine, Pampanga State Agricultural University, Pampanga 2011, Philippines

**Keywords:** PDCoV, recombinant S1 protein, ELISA, diagnostic performance, cross reactivity

## Abstract

Porcine deltacoronavirus (PDCoV), belonging to family *Coronaviridae* and genus *Deltacoronavirus*, is a major enteric pathogen in swine. Accurate PDCoV diagnosis relying on laboratory testing and antibody detection is an important approach. This study evaluated the potential of the receptor-binding subunit of the PDCoV spike protein (S1), generated using a mammalian expression system, for specific antibody detection via indirect enzyme-linked immunosorbent assay (ELISA). Serum samples were collected at day post-inoculation (DPI) −7 to 42, from pigs (*n* = 83) experimentally inoculated with different porcine coronaviruses (PorCoV). The diagnostic sensitivity of the PDCoV S1-based ELISA was evaluated using serum samples (*n* = 72) from PDCoV-inoculated animals. The diagnostic specificity and potential cross-reactivity of the assay was evaluated on PorCoV-negative samples (*n* = 345) and samples collected from pigs experimentally inoculated with other PorCoVs (*n* = 472). The overall diagnostic performance, time of detection, and detection rate over time varied across different S/P cut-offs, estimated by Receiver Operating Characteristic (ROC) curve analysis. The higher detection rate in the PDCoV group was observed after DPI 21. An S/P cut-off of 0.25 provided 100% specificity with no serological cross-reactivity against other PorCoV. These results support the use of S1 protein-based ELISA for accurate detection of PDCoV infections, transference of maternal antibodies, or active surveillance.

## 1. Introduction

Porcine deltacoronavirus (PDCoV) is an enveloped, singled-stranded, positive-sense RNA virus that belongs to the order *Nidovirales*, family *Coronaviridae*, subfamily *Coronavirinae*, genus *Deltacoronavirus*, subgenus *Buldecovirus*, and species coronavirus HKU15 [1]. Deltacoronavirus species evolved from a bird coronavirus (CoV) lineage of broad diversity, cross-infecting other bird species, and occasionally some mammalian species like pigs [2]. The PDCoV genome (~25.4 kb) consists of a 5′ and 3′ untranslated region (UTR) and eight open reading frames (ORFs), which are, in order, the replicase-transcriptase ORF1 (ORF1a and ORF1b), spike (S), envelope (E), membrane (M), accessory gene (NS) 6, nucleocapsid (N), NS7a, and NS7 [3,4,5,6].

PDCoV (originally “PorCoV HK15”) was first detected and identified by sequencing during a surveillance study carried out in Hong Kong in 2012 [2], but it was not until 2014 that PDCoV emerged in the USA (Ohio and Indiana) associated with clinical cases of diarrhea, vomiting, and dehydration in suckling pigs [7,8]. Subsequently, it spread quickly to other states in the USA [9] and Canada [10]. PDCoV has also been reported in South Korea [11], China [12], Thailand, Vietnam and Laos [13], Japan [14], Mexico [15], and Peru [16].

Besides PDCoV, there are at least four other enteropathogenic porcine coronaviruses (PorCoV) that have been described to date, which include transmissible gastroenteritis coronavirus (TGEV) [17], porcine epidemic diarrhea virus (PEDV) [18], a recombinant PEDV/TGEV coronavirus [19], and a swine acute diarrhea syndrome coronavirus (SADS-CoV) [20], all within the genus *Alphacoronavirus*. In addition, porcine hemagglutinating encephalomyelitis virus (PHEV) [21] in genus *Betacoronavirus*, and porcine respiratory coronavirus (PRCV) [22] in genus *Alphacoronavirus*, are commonly associated with neurological/digestive and respiratory diseases, respectively.

Like other enteropathogenic swine viruses, PDCoV infects villous enterocytes, causing atrophic enteritis that leads to malabsorptive diarrhea and significant mortality in suckling pigs as demonstrated through experimental studies [23,24,25,26] and field investigations [12,27,28,29]. Although generally less virulent than high pathogenic PEDV and TGEV strains, PDCoV has steadily gained clinical relevance in the field.

As reviewed by Zhang [30], different methods and techniques have been described for PDCoV detection in clinical specimens, including electron microscopy, virus isolation in vitro and in vivo assessment of virus viability, different polymerase chain reaction (PCR) methods, immunofluorescence, immunohistochemistry, and various antibody detection methods that were designed to aid in the diagnosis, characterization, and epidemiological investigations of PDCoV-associated enteric disease.

The fact that the histologic findings are non-pathognomonic, the non-specific clinical presentation of PDCoV-associated disease, and the co-circulation of different PorCoVs in commercial swine herds, make the differential diagnosis dependent on laboratory testing. In this study, we assessed the potential of a recombinant protein derived from the amino (N)-terminal receptor-binding subunit of the PDCoV S protein (S1) for specific detection of PDCoV antibodies and the presence or absence of serologic cross-reactivity against other PorCoV.

## 2. Results

The soluble expression of Fc-tag-fused S1 proteins (761 aa; 84.8 kDa; pI: 6.3) was confirmed by 12% SDS-PAGE (Figure 1), allowing its purification under native conditions. PDCoV S1-Fc protein was first purified from culture supernatant by protein A chromatography (Figure 2a). The eluted fractions of Fc-tag-fused PDCoV S1 proteins were pooled, dialyzed (PBS pH 3.0), and subjected to enzymatic cleavage (TEV 20 IU/mg sample) at 37 °C overnight (~16 h), under endotoxin control (Figure 2b). TEV-digested PDCoV S1 His-tagged protein (510 aa; 56.6 kDa; pI: 5.8) fractions were subjected to further purification by HisTrap^TM^ FF and protein A affinity columns to separate the cleaved Fc tag (Figure 2c). Then, a Superdex^®^ 200 exclusion chromatography column was used for further enrichment of the Fc-cleaved S1 protein, the selected fractions were pooled (~0.6 mg/mL), dialyzed against PBS pH 7.4, and analyzed by SDS-PAGE (Figure 2d).

This S1 recombinant protein was then used to develop an indirect (IgG) ELISA. The distribution of PDCoV S1-based ELISA IgG S/P values obtained for each inoculation group is shown in Figure 3. The ROC analysis performed on the cumulative data collected from PDCoV-positive and PDCoV-negative samples tested by the S1-based indirect ELISA showed that the diagnostic performance of the PDCoV S1-based ELISA was subjected to a specific cut-off (Figure 4). The use of an ELISA S/P cutoff of 0.25 ensured a specificity of 100%, without serologic cross-reactivity against other PorCoV, while a cutoff of 0.10 maximized the diagnostic specificity. Thus, based on the ROC analysis, the detection rate of seropositive animals over time was estimated at different S/P cutoffs (0.10, 018, and 0.25) (Table 1). The first positive animals were detected between DPI 7 (3/12), when using an S/P cutoff of 0.1, and DPI 10 using a cutoff of 0.18 (2/12) or 0.25 (3/12). The higher detection rate (12/12) was observed at DPI 42 with an S/P cutoff of 0.10 (Table 1). Variable seroconversion was observed across pigs regardless the cutoff. Only one pig over 12 (pig 9) was seronegative throughout the study when an S/P cutoff value of 0.25 was used for result interpretation.

## 3. Discussion

In the absence of commercial vaccines, the detection of PDCoV antibodies indicates that an animal is or was previously infected or received passive antibodies through lactation. Different immunoassays for the detection of PDCoV antibodies have been described [31,32,33,34,35,36,37]. However, pigs are exposed to different coronaviruses, and the potential antibody cross-reactivity may contribute to false-positive results, particularly when using conservative antigens for assay development [38,39,40]. Therefore, any immunoassays would have limited utility in the field unless the absence of potential cross-reactivity against other porcine coronaviruses commonly circulating in commercial swine herds is ruled out.

In a previous study focused on PEDV, we demonstrated that among different structural proteins evaluated, the N-terminal portion of the S protein (S1) not only contains major antigenic sites, but it is also virus-specific, showing no serological cross-reactivity with other porcine coronaviruses [39]. Immunoassays based on a more conservative protein (e.g., structural protein N) [23,31,33,38,41] are more susceptible to potential cross-reactivity with other coronaviruses. Therefore, among the different PDCoV structural and non-structural proteins, the present study evaluated the suitability of the S1 recombinant protein for specific detection of PDCoV antibodies used on an indirect ELISA platform.

As with other coronaviruses, PDCoV S protein is a glycosylated structural protein that consists of two subunits, the N-terminal S1 receptor-binding globular hydrophilic head and the C-terminal S2 membrane-fusion stalk, which is more conserved and highly hydrophobic [42]. Here, the S1 subunit was produced using a mammalian expression system and purified following a stepwise purification approach that combined affinity chromatography techniques, aided by the addition of both IgG-Fc and His tags, and molecular exclusion purification techniques. This strategy allowed for the generation of a recombinant protein in its native structure, including the appropriate post-translational modifications required for a glycosylated protein to be functional, e.g., proper antigen recognition, and receptor binding. Although the aminopeptidase N (APN) was initially proposed as a cellular receptor for PDCoV [43,44], further evidence suggested the existence of additional unknown receptors for PDCoV [45]. Different studies identified major APN receptor-binding domains in the PDCoV S1 protein that are key targets of neutralizing antibodies [46,47]. The neutralizing activity was partially attributed to the blockage of sugar-binding activity, highlighting the importance of glycosylation and native conformation of the S1 protein [47].

Previous reports described the development of PDCoV ELISAs, based on the S1 protein [32,36], but none of them adequately addressed the potential cross-reactivity with other PorCoVs. Thus, the main goal of this study was to assess the presence or absence of antibody cross-reactivity against anti-sera obtained from pigs individually infected under experimental conditions with different porcine coronaviruses currently circulating in most commercial herds. As expected, the diagnostic sensitivity and specificity, and the analytical specificity were directly associated with the specific cut-off used to interpret the ELISA results. No serological cross-reactivity was detected by ELISA when using an S/P cut-off value of 0.25, providing 100% specificity, though this had an impact on the overall time of detection and overall diagnostic sensitivity. Although all pigs were inoculated at the same time and with the same inoculum, both the time and the pattern of seroconversion varied across pigs. Contrarily, an S/P cutoff of 0.10 maximized the detection rate over time and overall diagnostic sensitivity but compromised the diagnostic specificity. In any case, the selection of an appropriate cutoff should be linked to the purpose of the testing and status of the herd (e.g., diagnosis vs. surveillance).

Previous studies in poultry [48], calves [49], mice [47], and even humans [50] evidenced the potential of PDCoV for interspecies and cross-species transmission. The PDCoV S1-based indirect ELISA described herein could be easily adapted for antibody detection in different animal species simply by switching the labelled secondary antibody, i.e., species-specific antibody or non-species-specific protein A or G for primary binding antibodies. This could help in future serological investigations of the potential circulation of PDCoV across species.

This study concluded that, despite the complexity of the *Coronaviridae* family, coronavirus-specific serological tools could be developed when the right antigen target (i.e., S1) and the testing platform are appropriately selected and designed.

## 4. Materials and Methods

### 4.1. Experimental Samples of Known Porcine Coronavirus Infection Status

The animal study was approved by the Iowa State University Office for Responsible Research and the Institutional Animal Care and Use Committee (IACUC #12-17-8658-S). A conventional wean-to-finish farm with no history of porcine coronavirus infections was selected as the source to procure the pigs used in this study. Seven-week-old pigs (*n* = 83) that tested negative for PDCoV, PEDV, TGEV, PRCV, and PHEV by quantitative reverse transcription PCR (RT-qPCR) [23,51,52,53] and serologic [immunofluorescence assay (IFA) or enzyme-linked immunosorbent assay (ELISA)] methods [23,39,40,52,54] were randomized into seven inoculation groups allocated to separate rooms. Detailed information about specific virus strains, virus titers, inoculum, inoculation routes, and number of pigs per group of inoculation is presented in Table 2. The pigs were closely observed twice daily for clinical signs throughout the study. Serum samples were collected from each group on day post-inoculation (DPI) –7, 0, 3, 7, 10, 14, 17, 21, 28, 35, and 42. All pigs were euthanized at DPI 42 by penetrative captive bolt (Accles and Shelvoke, Ltd., Sutton Coldfield, UK).

### 4.2. Generation of PDCoV S1 Recombinant Protein

With minor modifications, the PDCoV S1 recombinant protein was generated using a mammalian expression system as previously described for PEDV [39] and PHEV [54]. In brief, the coding region of the PDCoV S1 subunit gene was synthetically produced (Shanghai Genery Biotech Co., Ltd., Shanghai, China) with the addition of a 5′ terminal eukaryotic native signal, followed by a 3′ terminal Tobacco Etch Virus (TEV) cysteine protease site, and the Fc portion of human IgG1 (~2200 bp) (Table 3). After amplification by PCR, the amplicons were cloned into pNPM5 eukaryotic expression vector (Novoprotein, Short Hills, NJ, USA). Thereafter, human embryonic kidney (HEK) 293 cells (1 × 10^6^ cells/mL) (Invitrogen, Thermo Fisher Scientific, Grand Island, NY, USA) were transfected with the recombinant plasmid using polyethylenimine (PEI) (Thermo Fisher Scientific) at a 1:4 ratio (plasmid: PEI, *w*/*w*) for protein expression. The transfected cells were grown in serum-free FreeStyle™ 293 Expression Medium (Gibco, Life Technologies, Carlsbad, CA, USA) at 37 °C with 8% CO_2_ by orbital shaking at 130 rpm. Cells were harvested by centrifugation (3500× *g* for 20 min) on the fifth-day post-transfection, and culture supernatants were filter-sterilized (0.45 μm filter). Pilot assessment of protein expression was performed by dodecyl sulfate-polyacrylamide gel electrophoresis (SDS-PAGE), and protein purification was performed through affinity chromatography, i.e., protein A and HisTrap^TM^ FF and protein A affinity columns, according to manufacturer’s instructions (GE Healthcare, Pittsburgh, PA, USA). The last purification-polishing step was performed via a Superdex^®^ 200 exclusion chromatography column (GE Healthcare) for high-resolution separation of proteins according to size.

### 4.3. PDCoV Indirect ELISA

The optimum concentration of the purified PDCoV S1 recombinant protein for coating was determined via checkerboard titration using known antibody-positive and antibody-negative sera and maximizing the signal-to-noise ratio. Among the different coating concentrations evaluated (0.15, 0.3, 0.6, 1.2, and 2.4 μg/mL), the protein was diluted at 0.6 μg/mL in phosphate-buffered saline (PBS; Gibco^®^, Thermo Fisher Scientific, Grand Island, NY, USA) pH 7.4, coated (100 μL per well) onto 96-well plates (Universal Binding, catalog number 95029390; Thermo Fisher Scientific), and incubated at 4 °C for 16 h. Plate wells were washed five times with 350 μL PBS pH 7.4 containing 0.1% Tween 20 (PBST), blocked with a 1% (wt/vol) bovine serum albumin solution (Jackson ImmunoResearch Inc., West Grove, PA, USA), incubated at room temperature (20–25 °C) for 2 h, dried at 37 °C for 3 h, sealed and stored at 4 °C until use. Coated plates were subject to quality control (intra- and inter-plate; percent coefficient of variation ≤ 12%).

Serum samples (single well) and internal positive and negative controls (duplicate) were diluted to 1/100 in PBS pH 7.4 solution containing 50% goat serum (Gibco^®^, Thermo Fisher Scientific), and added to the PDCoV S1 protein-coated plates. The plates were incubated for 1 h at 37 °C, then washed three times with 350 μL per well of PBST. Next, 100 μL of HRP-labelled goat anti-pig IgG (Fc) (Bethyl Laboratories Inc., Montgomery, TX, USA) at 1:50,000 was added to each well, followed by a 30 min incubation at 37 °C. After an additional washing step with PBST, a 100 μL tetramethylbenzidine-hydrogen peroxide (TMB) substrate solution (SurModics IVD, Inc., Eden Prairie, MN, USA) was added to each well and incubated for 5 min at room temperature in the dark. The reaction was stopped by adding 100 μL of stop solution (SurModics IVD, Inc.) per well, and the optical density (OD_450_) at 450 nanometers was measured using an ELISA plate reader (SoftMax Pro 7; Molecular Devices, San Jose, CA, USA) operated with SoftMax Pro7 software (Molecular Devices). Antibody responses were expressed as the sample-to-positive (S/P) ratios:
$$S/P\;ratio=\frac{\left(\mathrm{sample}{\mathrm{OD}}_{450}-\mathrm{negative}\mathrm{control}\mathrm{mean}{\mathrm{OD}}_{450}\right)}{\left(\mathrm{positive}\mathrm{control}\mathrm{mean}{\mathrm{OD}}_{450}-\mathrm{negative}\mathrm{control}\mathrm{mean}{\mathrm{OD}}_{450}\right)}$$

### 4.4. Data Analysis

The ELISA S/P serum IgG responses across inoculation groups were plotted using GraphPad Prism^®^ (GraphPad Software Inc., La Jolla, CA, USA). The R software package pROC (https://cran.r-project.org/web/packages/pROC/index.html, accessed on 24 July 2022) [56] was used to perform a Receiver Operating Characteristic (ROC) analysis on experimental samples of known PorCoV infection status to estimate the diagnostic performance of the PDCoV S1-based indirect ELISA. The non-parametric DeLong method was used to estimate the 95% confidence intervals (CIs) for the area under curve (AUC) [57]. Before performing the analyses, the S/P data were normalized by a 1/3 power transformation; diagnostic sensitivity and specificity were derived from the ROC analyses for specific assay cut-offs.

The diagnostic sensitivity was estimated on sera from PDCoV-inoculated pigs collected after 14 DPI (*n* = 72). The diagnostic specificity and potential cross-reactivity of the PDCoV S1-based ELISA was assessed on serum samples from PorCoV-negative pigs (*n* = 333) and pigs inoculated with PEDV, TGEV, PRCV, and PHEV between DPI 7 to 42 (*n* = 472).

Typical methods for estimating confidence intervals of proportions are based on binomial distribution and do not account for the correlated structure of longitudinal data, i.e., repeated observations from the same animals over time. Therefore, a method for deriving CI for correlated, normally distributed data was utilized [39]. In brief, the correlation of the data was taken into account by fitting normalized data into a linear mixed model.
(1)$$Y_{ij}=µ+\gamma i+{\tau s}_{ij}+\epsilon_{ij}$$

In Equation (1), *Y_ij_* is the *j*^th^ observation for the *i*^th^ subject, *µ* is the overall mean for samples classified as PDCoV antibody-negative, *γ_i_* is the random effect of the *i*^th^ subject, *τ* is the fixed effect indicating the mean difference between PDCoV antibody-negative and positive groups; *s_ij_* is the disease status of the *j*^th^ observation for the *i*^th^ subject; and *ϵ_ij_* is the random error of the *j*^th^ observation for the *i*^th^ subject. The variances from equation 1 were used to calculate the 95% CIs for ROC-derived diagnostic sensitivity and diagnostic specificity estimates. Logit transformation was used to prevent the estimated intervals from exceeding the range of probability, i.e., (0, 1).

## Figures and Tables

**Figure 1 pathogens-11-00910-f001:**
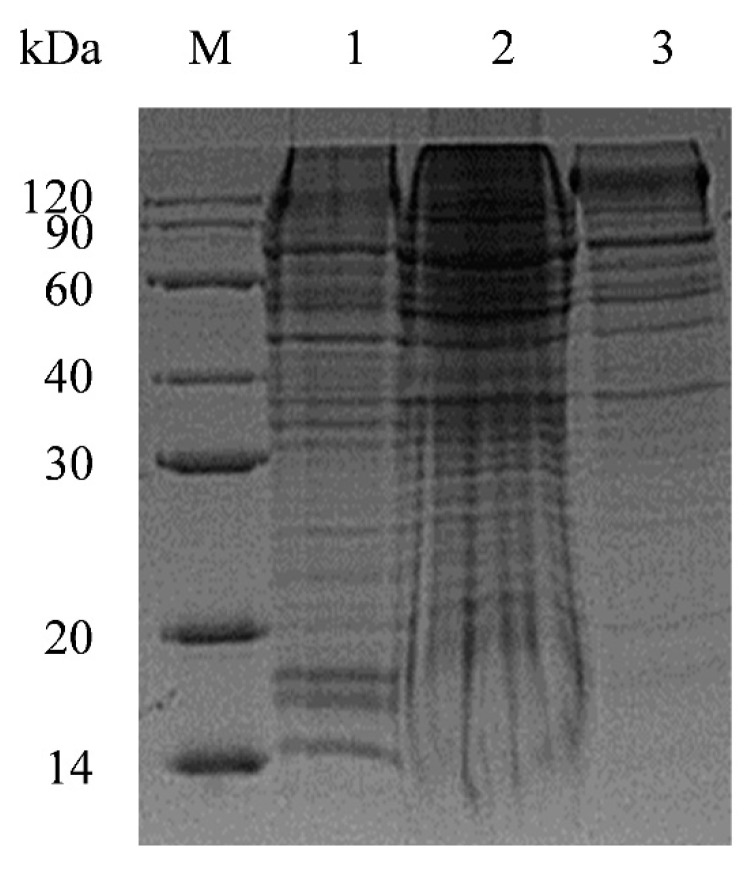
Analysis of porcine deltacoronavirus (PDCoV) S1-Fc-fused recombinant protein (~84.8 kDa) expression and secretion in culture medium using sodium dodecyl sulfate–polyacrylamide gel electrophoresis (SDS-PAGE). Lane M: molecular weight protein marker; Lane 1: supernatant of cell culture; Lane 2: supernatant of cell lysate; Lane 3: precipitate of cell lysate.

**Figure 2 pathogens-11-00910-f002:**
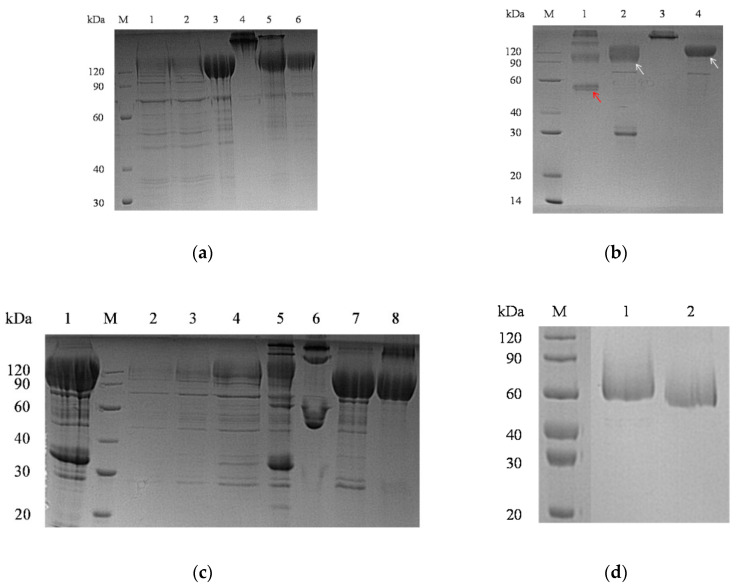
Analysis of porcine deltacoronavirus (PDCoV) S1-Fc-fused recombinant protein purification via protein A chromatography (GE Healthcare) from HEK293 cell culture supernatant using sodium dodecyl sulfate–polyacrylamide gel electrophoresis (SDS-PAGE) (**a**) Lane M: molecular weight protein marker; Lanes 1–2: non-bound protein flow-through; Lanes 3 and 5: elution by 0.1 M glycine, pH 3.0 (reduced); Lane 4: elution by 0.1 M glycine, pH 3.0 (non-reduced); Lane 6: elution by 0.1 M glycine, pH 2.5 (reduced). Eluted proteins corresponding to lanes 3, 5, and 6, were pooled and dialyzed to phosphate-buffered saline pH 3.0 for next step Tobacco Etch Virus (TEV) cysteine protease enzyme digestion (**b**) Lane M: molecular weight protein marker; Lane 1: PDCoV S1 protein after TEV-cleavage (reduced; red arrow); Lane 2: PDCoV S1 protein after TEV-cleavage (non-reduced; white arrow); Lane 3: PDCoV S1 protein before TEV-cleavage (non-reduced); Lane 4: PDCoV S1 protein before TEV-cleavage (reduced; white arrow). Eluted fractions after TEV-cleavage were subjected to further purification by HisTrap^TM^ FF and protein A affinity columns (GE Healthcare, Chicago, IL, USA) to separate the cleaved Fc tag (**c**) Lane 1: PDCoV S1-Fc fused protein; M: molecular weight protein marker; Lane 2: load sample (column equilibrated with 20 mM phosphate buffer, 500 mM NaCl, pH 7.4); Lane 3–4: flow-through; Lane 5–6: elution by 0.1 M glycine pH 2.5; Lane 7: elution by 0.1 M glycine pH 2.5 (non-reduced); Lane 8: elution by 0.5 M glycine pH 2.5. Lane 7 protein was subjected for next step purification through exclusion chromatography (Superdex^®^ 200; GE Healthcare), and selected fractions were dialyzed against phosphate buffered saline pH 7.4 as final product (**d**) M: molecular weight protein marker; Lane 1: PDCoV S1 protein (~55.8 kDa) (reduced); Lane 2: S1-PDCoV (non-reduced).

**Figure 3 pathogens-11-00910-f003:**
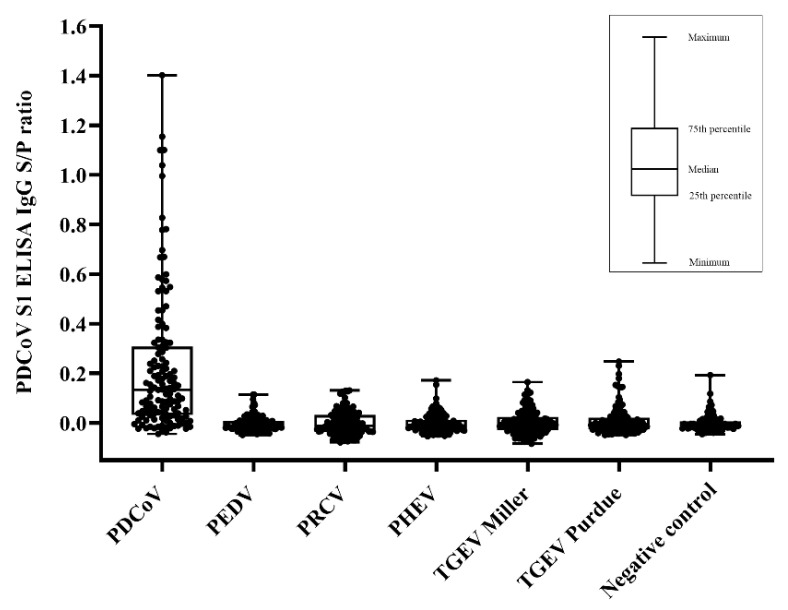
Distribution of cumulative porcine deltacoronavirus (PDCoV) S1-based IgG ELISA sample-to-positive (S/P) ratios from serum samples collected at day post-inoculation (DPI) −7 to 42 from pigs inoculated with different porcine coronaviruses (PorCoVs) under experimental conditions. Each dot represents a sample from the different PorCoVs inoculation groups including PDCoV (*n* = 132), porcine epidemic diarrhea virus (PEDV) (*n* = 132), porcine respiratory coronavirus (PRCV) (*n* = 132), porcine hemagglutinating encephalomyelitis virus (PHEV) (*n* = 132), transmissible gastroenteritis virus (TGEV) strains Miller (*n* = 132) and Purdue (*n* = 121), and negative control (*n* = 132) groups.

**Figure 4 pathogens-11-00910-f004:**
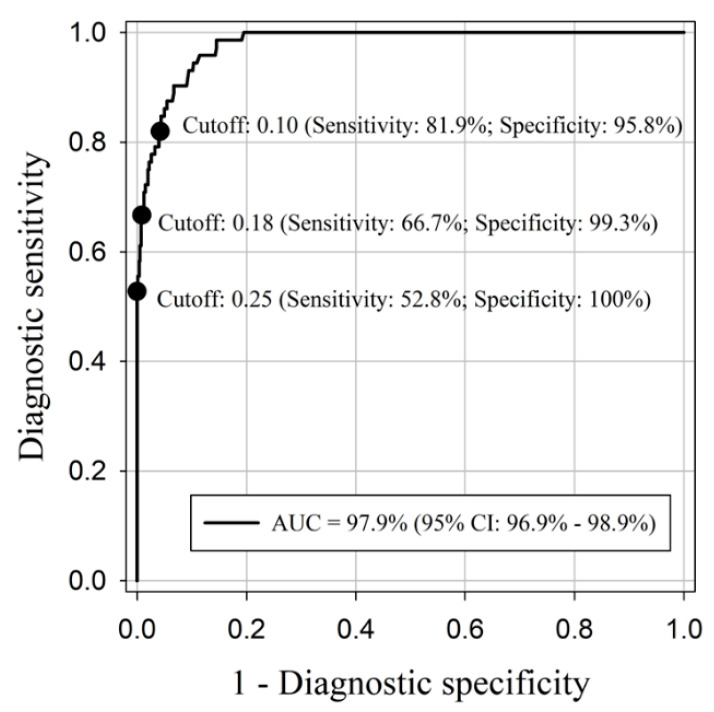
Receiver operating characteristic (ROC) analysis of the porcine deltacoronavirus (PDCoV) S1-based ELISA, showing diagnostic sensitivity (%) and specificity (%) by cutoffs.

**Table 1 pathogens-11-00910-t001:** Antibody (IgG) detection rate over time via porcine deltacoronavirus S1-based ELISA testing after experimental oral inoculation with PDCoV (USA/IL/2014 strain), using different sample-to-positive (S/P) cut-off values (0.10, 0.18, and 0.25). The table contains both individual (pig) and total detection by day post-inoculation. Samples tested positive for PDCoV antibodies are shown in red. Grey shaded boxes indicate changes in the qualitative results (negative to positive) at different S/P cutoff values.

Pig ID	Cutoff	Day Post-Inoculation							
		7	10	14	17	21	28	35	42
1	0.25	Neg ^a^	Neg	Pos	Neg	Pos	Pos	Pos	Pos
	0.18	Neg	Neg	Pos	Pos	Pos	Pos	Pos	Pos
	0.10	Neg	Pos ^b^	Pos	Pos	Pos	Pos	Pos	Pos
2	0.25	Neg	Neg	Pos	Pos	Pos	Pos	Pos	Pos
	0.18	Neg	Pos	Pos	Pos	Pos	Pos	Pos	Pos
	0.10	Neg	Pos	Pos	Pos	Pos	Pos	Pos	Pos
3	0.25	Neg	Pos	Neg	Pos	Pos	Pos	Neg	Pos
	0.18	Neg	Pos	Neg	Pos	Pos	Pos	Neg	Pos
	0.10	Pos	Pos	Pos	Pos	Pos	Pos	Neg	Pos
4	0.25	Neg	Neg	Pos	Pos	Pos	Pos	Neg	Pos
	0.18	Neg	Neg	Pos	Pos	Pos	Pos	Pos	Pos
	0.10	Pos	Neg	Pos	Pos	Pos	Pos	Pos	Pos
5	0.25	Neg	Neg	Neg	Pos	Neg	Neg	Pos	Neg
	0.18	Neg	Neg	Neg	Pos	Neg	Neg	Pos	Pos
	0.10	Neg	Neg	Neg	Pos	Neg	Neg	Pos	Pos
6	0.25	Neg	Neg	Neg	Neg	Pos	Pos	Neg	Pos
	0.18	Neg	Neg	Neg	Pos	Pos	Pos	Neg	Pos
	0.10	Neg	Neg	Pos	Pos	Pos	Pos	Neg	Pos
7	0.25	Neg	Neg	Pos	Pos	Pos	Pos	Neg	Pos
	0.18	Neg	Neg	Pos	Pos	Pos	Pos	Neg	Pos
	0.10	Neg	Neg	Pos	Pos	Pos	Pos	Neg	Pos
8	0.25	Neg	Neg	Neg	Neg	Pos	Neg	Pos	Neg
	0.18	Neg	Neg	Pos	Neg	Pos	Neg	Pos	Neg
	0.10	Neg	Neg	Pos	Neg	Pos	Pos	Pos	Pos
9	0.25	Neg	Neg	Neg	Neg	Neg	Neg	Neg	Neg
	0.18	Neg	Neg	Neg	Pos	Neg	Neg	Neg	Neg
	0.10	Neg	Neg	Neg	Pos	Neg	Neg	Pos	Pos
10	0.25	Neg	Neg	Neg	Pos	Neg	Neg	Pos	Neg
	0.18	Neg	Neg	Pos	Pos	Neg	Pos	Pos	Neg
	0.10	Neg	Pos	Pos	Pos	Pos	Pos	Pos	Pos
11	0.25	Neg	Neg	Neg	Neg	Neg	Pos	Pos	Pos
	0.18	Neg	Neg	Neg	Neg	Neg	Pos	Pos	Pos
	0.10	Neg	Neg	Neg	Pos	Pos	Pos	Pos	Pos
12	0.25	Neg	Neg	Neg	Neg	Neg	Neg	Pos	Neg
	0.18	Neg	Neg	Neg	Neg	Neg	Pos	Pos	Pos
	0.10	Pos	Neg	Neg	Pos	Neg	Pos	Pos	Pos
Total	0.25	0/12 (0%)	1/12 (8.3%)	4/12 (33.3%)	6/12 (50%)	7/12 (58.3%)	7/12 (58.3%)	7/12 (58.3%)	7/12 (58.3%)
	0.18	0/12 (0%)	2/12 (16.7%)	6/12 (50%)	9/12 (75%)	7/12 (58.3%)	9/12 (75%)	8/12 (66.7%)	9/12 (75%)
	0.10	3/12 (25%)	4/12 (33.3%)	8/12 (66.7%)	11/12 (91.7%)	9/12 (75%)	10/12 (83.3%)	9/12 (75%)	12/12 (100%)

^a^ Antibody negative; ^b^ Antibody Positive.

**Table 2 pathogens-11-00910-t002:** Porcine coronaviruses, inoculum dose and route of inoculation used for experimental inoculations.

	Virus Strain	Cell Line	Virus Titer (TCID_50_/mL)	Virus Inoculum (mL)	Inoculation Route	Reference
PDCoV (12)	USA/IL/2014	Swine testicle (ATCC^®^ CRL-1746)	1.5 × 10^6^	30	Orogastric	[23]
TGEV Miller (12)	ATCC ^a^ VR-1740		4.0 × 10^6^	35	Orogastric	[39,55]
TGEV Purdue (12)	ATCC VR-763		2.4 × 10^8^	30	Orogastric	
PRCV (12)	ATCC VR-2384		4.0 × 10^5^	15	Nasal	
PEDV (12)	USA/IN/2013/19338E	Vero (ATCC^®^ CCL-81)	1.5 × 10^6^	15	Orogastric	
PHEV (12)	NVSL PHEV 67N	Swine kidney primary cells (NVSL-USDA ^b^)	1:128 ^c^	5	Oronasal	[53]
Negative control (12)	Culture medium	-	-	20	Oronasal	-

^a^ ATCC: American Type Culture Collection; ^b^ NVSL: National Veterinary Service Laboratory—United States Department of Agriculture; ^c^ Hemagglutination titer.

**Table 3 pathogens-11-00910-t003:** Primers, DNA and amino acid sequences of the codon-optimized PDCoV S1 protein for cloning and expression using a mammalian expression system.

	Sequences
DNA	ATGACATCCACTTTGCCTTTCTCTCCACAGGTGTCCACTCCCAGGTCCAAGTTTAAACGGATCTCTAGCGAATTCGCCGCCACCATGCAGAGAGCACTGCTGATTATGACTCTGCTGTGTCTGGTCAGAGCTAAGTTCGCTGATGATCTGCTGGACCTGCTGACATTCCCTGGAGCTCATAGATTCCTGCATAAGCCTACCAGGAACAGCAGCTCCCTGTATTCCAGGGCTAACAACAACTTCGATGTGGGAGTGCTGCCTGGATACCCTACCAAGAACGTCAACCTGTTTAGCCCTCTGACAAATTCCACCCTGCCCATCAACGGACTGCACAGAAGCTACCAGCCTCTGATGCTGAATTGCCTGACTAAGATTACCAACCACACCCTGAGCATGTACCTGCTGCCCTCCGAAATCCAGACCTACAGCTGCGGAGGCGCCATGGTCAAATACCAAACTCATGATGCAGTGAGGATCATCCTGGATCTGACTGCCACAGACCACATCTCCGTCGAAGTGGTCGGCCAGCACGGAGAGAACTACGTGTTTGTGTGTAGCGAGCAGTTTAACTACACCACCGCCCTGCACAATAGCACATTCTTCAGCCTGAACTCCGAACTGTACTGCTTCACCAACAACACATACCTGGGCATCCTGCCACCCGACCTGACCGACTTCACTGTCTACAGGACCGGCCAGTTCTACGCCAATGGCTATCTGCTGGGAACACTGCCTATTACCGTGAACTATGTGAGACTGTATAGAGGCCACCTGAGCGCCAACAGCGCCCACTTTGCTCTGGCCAATCTGACAGATACTCTGATCACACTGACCAACACAACTATCAGCCAGATTACATACTGCGACAAGAGCGTGGTGGACAGCATCGCCTGCCAGAGAAGCAGCCACGAGGTGGAGGACGGCTTCTACTCCGATCCCAAATCCGCCGTCAGGGCAAGACAAAGGACTATCGTCACTCTGCCCAAGCTGCCCGAGCTGGAGGTCGTGCAGCTGAACATTTCCGCCCACATGGACTTCGGAGAAGCCAGGCTGGATAGCGTGACCATCAATGGCAACACCAGCTATTGCGTGACAAAGCCTTACTTCAGACTGGAGACAAACTTCATGTGCACCGGCTGCACCATGAACCTGAGGACCGACACCTGCAGCTTCGATCTGTCCGCTGTCAACAACGGGATGTCCTTCTCCCAATTCTGTCTGAGCACCGAGTCCGGAGCATGCGAGATGAAGATCATTGTGACCTACGTCTGGAATTACCTGCTGAGGCAGAGGCTGTATGTCACTGCCGTGGAAGGCCAAACCCACACCGGAACCACCTCCGTGCATGCCACTGACACTAGCTCCGTCATCACTGATGTGTGCACTGATTACACCATCTACGGCGTGAGCGGCACCGGGATCATTAAGCCCAGCGATCTGCTGCTGCACAACGGCATCGCTTTCACCTCTCCCACCGGCGAGCTGTACGCCTTCAAGAATATCACTACCGGCAAGACCCTGCAAGTCCTGCCTTGCGAGACACCCAGCCAGCTGATTGTCATCAACAATACCGTCGTGGGAGCAATCACAAGCTCCAACTCCACCGAGAACAATAGGTTCACCACAACAATCGTGACACCAACCTTCTTCTACGAGAACCTGTACTTCCAGAGCGGCTCCGACAAGACCCACACCGTCGAGTGCCCACCGTGCCCAGCACCTGAACTCCTGGGGGGACCGTCAGTCTTCCTCTTCCCCCCAAAACCCAAGGACACCCTCATGATCTCCCGGACCCCTGAGGTCACATGCGTGGTGGTGGACGTGAGCCACGAAGACCCTGAGGTCAAGTTCAACTGGTACGTGGACGGCGTGGAGGTGCATAATGCCAAGACAAAGCCGCGGGAGGAGCAGTACAACAGCACGTACCGTGTGGTCAGCGTCCTCACCGTCCTGCACCAGGACTGGCTGAATGGCAAGGAGTACAAGTGCAAGGTCTCCAACAAAGCCCTCCCAGCCCCCATCGAGAAAACCATCTCCAAAGCCAAAGGGCAGCCCCGAGAACCACAGGTGTACACCCTGCCCCCATCCCGGGAGGAGATGACCAAGAACCAGGTCAGCCTGACCTGCCTGGTCAAAGGCTTCTATCCCAGCGACATCGCCGTGGAGTGGGAGAGCAATGGGCAGCCGGAGAACAACTACAAGACCACGCCTCCCGTGCTGGACTCCGACGGCTCCTTCTTCCTCTATAGCAAGCTCACCGTGGACAAGAGCAGGTGGCAGCAGGGGAACGTCTTCTCATGCTCCGTGATGCATGAGGCTCTGCACAACCACTACACGCAGAAGAGCCTCTCCCTGTCTCCGGGTAAATGA
Primers	SPDCV-F-F: 5′-TAAACGGATCTCTAGCGAATTCGCCGCCACCATGCAGAGAGC-3′ SPDCV-F-R1:5′-GTCTTGTCGGAGCCGCTCTGGAAGTACAGGTTCTCGTAGAAGAAGGTTGGTGTCAC-3′ SPEDV-F1: 5′-CTTCCAGAGCGGCTCCGACAAGACCCACACCGTCGAGTGCCCACCGTGCCCAG-3′ SPEDV-R: 5′-CGAGCGGCCGCTAGCAAGCTTTCATTTACCCGGAGACAGGGAG-3′
Amino acid	MQRALLIMTLLCLVRAKFADDLLDLLTFPGAHRFLHKPTRNSSSLYSRANNNFDVGVLPGYPTKNVNLFSPLTNSTLPINGLHRSYQPLMLNCLTKITNHTLSMYLLPSEIQTYSCGGAMVKYQTHDAVRIILDLTATDHISVEVVGQHGENYVFVCSEQFNYTTALHNSTFFSLNSELYCFTNNTYLGILPPDLTDFTVYRTGQFYANGYLLGTLPITVNYVRLYRGHLSANSAHFALANLTDTLITLTNTTISQITYCDKSVVDSIACQRSSHEVEDGFYSDPKSAVRARQRTIVTLPKLPELEVVQLNISAHMDFGEARLDSVTINGNTSYCVTKPYFRLETNFMCTGCTMNLRTDTCSFDLSAVNNGMSFSQFCLSTESGACEMKIIVTYVWNYLLRQRLYVTAVEGQTHTGTTSVHATDTSSVITDVCTDYTIYGVSGTGIIKPSDLLLHNGIAFTSPTGELYAFKNITTGKTLQVLPCETPSQLIVINNTVVGAITSSNSTENNRFTTTIVTPTFFYENLYFQSGSDKTHTVECPPCPAPELLGGPSVFLFPPKPKDTLMISRTPEVTCVVVDVSHEDPEVKFNWYVDGVEVHNAKTKPREEQYNSTYRVVSVLTVLHQDWLNGKEYKCKVSNKALPAPIEKTISKAKGQPREPQVYTLPPSREEMTKNQVSLTCLVKGFYPSDIAVEWESNGQPENNYKTTPPVLDSDGSFFLYSKLTVDKSRWQQGNVFSCSVMHEALHNHYTQKSLSLSPGK

Signal peptide (blue); Tobacco Etch Virus (TEV) cysteine protease site (green); Human IgG1 Fc tag (purple).

## Data Availability

All data are available from the corresponding author upon reasonable request.

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
