# Peer review of "The N-terminal Subunit of the Porcine Deltacoronavirus Spike Recombinant Protein (S1) Does Not Serologically Cross-react with Other Porcine Coronaviruses"

_pathogens, 2022, doi:10.3390/pathogens11080910_

Round 1
Author Response
August 8, 2022
Re: pathogens-1856738 (The N-terminal subunit of the porcine deltacoronavirus spike recombinant protein (S1) does not serologically cross-react with other porcine coronaviruses)
Response to Reviewer #1:
Dear Reviewer,
We thank you for your comments and suggestions. We feel they have resulted in an improved manuscript and appreciate your efforts on our behalf. Line numbers listed below refer to the edited manuscript with "no markup" option turned on.
Sincerely,
Luis G. Giménez-Lirola, BS, MS, PhD
Minor revision:
- As PDCoV S1 protein was used in indirect enzyme-linked immunosorbent assay, which can be used in diagnosis, please evaluate the sensitivity of S1-base ELISA.
Response: The manuscript has been extensively reviewed to address this concern. Specifically, we have modified:
- Fig. 4, to show the overall diagnostic sensitivity and specificity at different S/P cutoff values, selected from ROC analysis.
- Table 1, to show the total detection rate by DPI at different S/P cutoffs.
- The result (Line 119-127) and discussion (Line 202-205) sections to add additional information on the diagnostic performance.
- Also, please asses the diverse concentrations of PDCoV S1 protein used in ELISA, to ensure what concentration is the best.
Response: The M&M ELISA section of the manuscript has been reviewed to include this information (Line 268-271).

Reviewer 2 Report
The authors developed an indirect enzyme-linked immunosorbent assay (ELISA) using a recombinant protein derived from the amino (N)-terminal receptor-binding subunit of the Porcine deltacoronavirus (PDCoV) S protein (S1) for specific detection of PDCoV antibody. Moreover, the authors evaluated the serologic cross-reaction against other porcine coronaviruses (PorCoV).
Minor revision:
Page 9, line 244: the authors must inform the type of 96-well plate that was used in the indirect ELISA, for example Maxisorp, Medisorp, etc.
Page 10, line 249: the authors must inform whether the samples were tested in replicates or not.
Author Response
August 8, 2022
Re: pathogens-1856738 (The N-terminal subunit of the porcine deltacoronavirus spike recombinant protein (S1) does not serologically cross-react with other porcine coronaviruses)
Response to Reviewer #2:
Dear Reviewer,
We thank you for your comments and suggestions. We feel they have resulted in an improved manuscript and appreciate your efforts on our behalf. Line numbers listed below refer to the edited manuscript with "no markup" option turned on.
Sincerely,
Luis G. Giménez-Lirola, BS, MS, PhD
Minor revision:
- Page 9, line 244: The authors must inform the type of 96-well plate that was used in the direct ELISA, for example Maxisorp, Medisorp, etc.
Response: Universal Binding (UB) flat bottom plates from Thermo Fisher Scientific (Cat#95029390) were used. The M&M section of the ELISA has been reviewed to add this specific information (Line 273-274).
- Page 10, line 249: the authors must inform whether the samples were tested in replicate or not.
Response: Once antigen coated, the ELISA plates were subject to quality control (repetitivity intra/inter-plate), using a % CV ≤ 12% for acceptance. Therefore, testing samples in duplicate was not necessary. Only controls were tested in duplicate. The ELISA procedure has been reviewed for clarity (Line 278-279).
